# Antimicrobial Resistance, Serologic and Molecular Characterization of *E. coli* Isolated from Calves with Severe or Fatal Enteritis in Bavaria, Germany

**DOI:** 10.3390/antibiotics11010023

**Published:** 2021-12-27

**Authors:** Andrea Feuerstein, Nelly Scuda, Corinna Klose, Angelika Hoffmann, Alexander Melchner, Kerstin Boll, Anna Rettinger, Shari Fell, Reinhard K. Straubinger, Julia M. Riehm

**Affiliations:** 1Bavarian Health and Food Safety Authority, 91058 Erlangen, Germany; heubeck.a95@gmail.com (A.F.); Nelly.Scuda@lgl.bayern.de (N.S.); Corinna.klose@lgl.bayern.de (C.K.); Angelika.Hoffmann@lgl.bayern.de (A.H.); Kerstin.Boll@lgl.bayern.de (K.B.); anna.Rettinger@lgl.bayern.de (A.R.); 2Bavarian Health and Food Safety Authority, 85764 Oberschleissheim, Germany; alexander.melchner@t-online.de; 3Chemical and Veterinary Investigation Office, 72488 Sigmaringen, Germany; Shari.Fell@web.de; 4Department of Veterinary Sciences, Faculty of Veterinary Medicine, Institute of Infectious Diseases and Zoonoses, Ludwig-Maximilians-University, 80539 Munich, Germany; R.Straubinger@lmu.de

**Keywords:** *E. coli*, calves, enteritis, antimicrobial resistance, serotypes, virulence, multidrug-resistant, extensively drug-resistant

## Abstract

Worldwide, enterotoxigenic *Escherichia coli* (ETEC) cause neonatal diarrhea and high mortality rates in newborn calves, leading to great economic losses. In Bavaria, Germany, no recent facts are available regarding the prevalence of virulence factors or antimicrobial resistance of ETEC in calves. Antimicrobial susceptibility of 8713 *E. coli* isolates obtained from 7358 samples of diseased or deceased diarrheic calves were investigated between 2015 to 2019. Considerably high rates of 84.2% multidrug-resistant and 15.8% extensively drug-resistant isolates were detected. The resistance situation of the first, second and third line antimicrobials for the treatment, here amoxicillin-clavulanate, enrofloxacin and trimethoprim-sulfamethoxazole, is currently acceptable with mean non-susceptibility rates of 28.1%, 37.9% and 50.0% over the investigated 5-year period. Furthermore, the ETEC serotypes O101:K28, O9:K35, O101:K30, O101:K32, O78:K80, O139:K82, O8:K87, O141:K85 and O147:K89, as well as the virulence factors F17, F41, F5, ST-I and stx1 were identified in a subset of samples collected in 2019 and 2020. The substantially high rates of multi- and extensively drug-resistant isolates underline the necessity of continuous monitoring regarding antimicrobial resistance to provide reliable prognoses and adjust recommendations for the treatment of bacterial infections in animals.

## 1. Introduction

*Escherichia coli* account to the major enteric and systemic pathogens of the Gram-negative rods within the family Enterobacteriaceae. Most of the *E. coli* colonizing the intestinal tract of animals and humans are commensal, but facultative pathogenic strains may cause intestinal disorder or even severe and life-threatening extraintestinal disease [1,2]. In calves, enterotoxigenic *E. coli* (ETEC) pose a leading cause of intestinal disease, especially within the first four days of life [3,4,5]. ETEC encode lipopolysaccharide structures (LPS) that may act as endotoxins, fimbrial adhesins and finally enterotoxins. The endotoxins within the blood stream cause fever, damage of endothelial cells and disseminated intravascular coagulation (DIC), that leads to acute shock and sudden death [1]. The serological LPS characterization in calves comprise the *E. coli* serogroups O8, O9 and O101, and respective serotypes O9:K35 and O101:K30, as these are known for endotoxin effect [6]. Further, the serotype O78:K80 plays a major role in systemic disease, septicemia and endotoxic shock of newborn calves [1,6,7]. In piglets, the serotype O141:K85 in combination with F4 fimbria is specific for the postweaning diarrhea syndrome [6]. As well, three further serotypes O139:K82, O8:K87 and O147:K89 play an important role as pathogens for swine [6,8]. Proteinaceous fimbrial adhesins precipitate the bacterial attachment to the enteric mucosa that avert the mechanical shedding of virulent strains from the gut by peristalsis [1,4,9]. Former studies showed that the fimbrial adhesins F5, F17 and F41 are associated with calf diarrhea [4]. For ETEC, two different types of enterotoxins contribute to diarrhea in calves, the heat-stable toxin (ST) and heat-labile toxin (LT), respectively [1,10,11]. On a molecular level, the toxins increase the second messengers cyclic adenosine/ guanosine monophosphate (cAMP/cGMP), that effect an active secretion of fluid and electrolytes in the small intestine leading to extreme loss of fluid within the organism [11,12]. Further, ruminants are known to be a major reservoir of human pathogenic Shiga toxin-producing *E. coli* (STEC) [13,14,15,16]. Shiga toxins (stx1, stx2) may lead to enterocyte damage, subsequent bloody diarrhea and endothelial damage leading to internal hemorrhages and septicemia in susceptible neonatal calves [1,17,18]. Enterohemorrhagic *E. coli* (EHEC), a subset of STEC, further include intimin, an adhesin coded from the enterocyte effacement pathogenicity island (eaeA) [19,20] and enterohemolysin, a toxin encoded by the ehxA gene [21]. As published in several case reports, a majority of human EHEC disease outbreaks are caused by the serotype O157:H7 originating from contaminated ground beef [13,22,23]. This serotype is responsible for the hemorrhagic colitis and the life-threatening hemolytic uremic syndrome with the occurrence of thrombocytopenia, hemolytic anemia and thrombotic microangiopathy that may lead to acute renal failure and death [23,24,25,26].

Worldwide, neonatal diarrhea is still a major economic problem on cattle farms and the therapy with antimicrobials is crucial in routine practice [27]. However, the medication with bactericide antibiotics is solely, but highly indicated exclusively in the case of life-threatening sepsis [28,29]. The Swiss antibiotic therapy guidelines for veterinarians recommend amoxicillin-clavulanate as a first line, sulfonamide-trimethoprim as a second line and fluoroquinolones as a third line choice, here enrofloxacin [29]. A study from 2014 revealed that veterinarians in Europe mainly used polymyxins (44%), (fluoro)quinolones (18%), penicillins (13%), aminoglycosides (9%) and third and fourth generation cephalosporins (8%) in calves with diarrhea emphasizing the problem of an inappropriate use of antibiotics [30]. This contributes to a higher level of antimicrobial resistant bacteria in young animals compared to adults [31,32,33]. In addition, the emergence of multidrug- and pandrug-resistant *E. coli* in fecal samples of diarrheic calves has been recently and repeatedly reported [33,34]. According to the expert proposal for standard definitions for acquired resistance from the European Centre for Disease Prevention and Control (ECDC), strains are classified as “multidrug-resistant” if these are non-susceptible (resistant or intermediate) to at least one antimicrobial agent in more than three categories. Isolates meet the definition “extensively drug-resistant” if these are non-susceptible in all agents but two or fewer categories. Finally, isolates non-susceptible to all agents in all antimicrobial categories are ranked as “pandrug-resistant” [35]. 

Previous data show that the prevalence of extended-spectrum β-lactamase (ESBL)-producing *E. coli* in calves increased from 7% to 29% between 2006 and 2013 in Germany [27]. ESBL-producing strains do encode for numerous resistance genes and may transduce these to other, even commensal, bacteria [36]. Animals hosting these *E. coli* bacteria constitute a resistance gene reservoir that may affect the health of man and animals [36,37]. 

Only few data are available on the identification of ETEC from calves in Bavaria. However, the discrimination between the physiological intestinal flora and pathogenic *E. coli* is crucial [1,6,38]. The aim of the present study was to provide recent information about the most prevalent pathotypes of *E. coli*. These include the investigation of the current virulence factors, serotypes and trends in antimicrobial resistance [9,39,40,41,42].

## 2. Results

### 2.1. Antimicrobial Susceptibility

Within the study period 8713 *E. coli* were isolated from 7358 diarrheic calves at the federal state veterinary laboratory in Bavaria, Germany (Appendix A). This number matches an average count of 1740 isolates per year that is in accordance with previous years (data not shown). The results on antimicrobial susceptibility testing revealed mean non-susceptibility values of 28.1% for amoxicillin-clavulanate, 37.9% for enrofloxacin and 50% for trimethoprim-sulfamethoxazole (Figure 1 and Figure 2, Appendix A). The highest non-susceptibility value of a substance within each antimicrobial class revealed 11.9% for tulathromycin (macrolides), 18.3% for colistin (polymixins), 61.9% for tetracycline (tetracyclines), 62.2% for spectinomycin (aminoglycosides), 69.7% for ampicillin (penicillins), 80.5% for cephalothin (cephalosporins) and 96.8 % for florfenicol (phenicols) (Figure 1). A 5-year tendency from 2015 to 2019, evaluated for amoxicillin-clavulanate, enrofloxacin and trimethoprim-sulfamethoxazole, revealed a statistically significant decrease of the non-susceptibility rates for amoxicillin-clavulanate and enrofloxacin (*p* < 0.05) (Figure 2, Table 1). Regarding trimethoprim-sulfamethoxazole a significant decrease was assessed from 51.9% to 47.8% between 2015 and 2017 regarding the non-susceptible *E. coli* isolates (*p* < 0.05). A subsequent increase was further revealed from 47.8% to 52.5% in the years 2017 to 2019 (*p* < 0.05) (Figure 2, Table 1). Categorizing the 8713 isolates according to the ECDC expert proposal, 84.2% of the isolates (7336/8713) were multidrug-resistant, 15.7% (1368/8713) were extensively drug-resistant, eight isolates (0.1%) were pandrug-resistant and one isolate was susceptible to all antimicrobials tested. As we only tested antimicrobials licensed for the veterinary use, and none of the latest antimicrobials available on the market, we rededicated the eight presumably pandrug-resistant as extensively drug-resistant summing up to 1376 isolates in this specification (Figure 3).

### 2.2. Serologic Characterization

Serotyping of a randomly chosen subset of 108 *E. coli* isolated in 2019 and 2020 revealed 38 unequivocally typeable (35.2%), 29 untypeable (26.8%) and 41 seronegative (38%) strains (Table 2, Appendix A). The most frequently detected serotypes were O101:K28 (8.3%; *n* = 9), O9:K35 and O139:K82 (6.5%; *n* = 7), O101:K30 (3.7%; *n* = 4), O101:K32, O78:K80 and O8:K87 (2.8%; *n* = 3). The serotypes O141:K85 and O147:K89 were detected once each (Table 2, Appendix A). Finally, the serotypes O138:K81, O149:K91 and O157:H7 were not detected at all.

The fimbrial antigen F5 agglutinated in 6.5% of the isolates (*n* = 7) in combination with the serotypes O101:K30, O101:K28 and O9:K35. The fimbrial antigen F4 agglutinated in 4.6% of the isolates (*n* = 5), and exclusively combined with the serotype O139:K82 (Table 2, Appendix A).

### 2.3. Molecular Characterization

Within the molecular characterization, 14 PCR assays targeted genes for the expression of fimbria, adhesin, hemolysin and toxins. A positive result was obtained for 24 isolates and 35 single assays, respectively (Table 2, Appendix A). The most frequently detected genes coded for the fimbria F17 (13.9%; 15/108), F41 (3.7%; 4/108) and F5. The latter was always detected in combination with the toxin gene coding for ST-I (6.5%; 7/108). Finally, the gene coding for stx1 was detected in two of 108 isolates (1.9%). Seven of 108 isolates (6.5%) carried more than one type of virulence-associated genes (Table 2, Appendix A). The fimbrial antigens F4, F6, F18, O157, adhesin eaeA, hemolysin ehxA and the toxins LT, ST-II and stx2 were not detected in any isolate. The occurrence of F4 fimbria in the serotyping assays could not be confirmed in the PCR investigation (Table 2, Appendix A). In all, 84 of 108 isolates were negative in all PCR assays (Table 2, Appendix A).

## 3. Discussion

Antibiotic treatment is the fundamental therapy regarding serious or life-threatening bacterial infections in man and animals [28,29]. Records regarding antimicrobial susceptibility on single substances are collected in many countries all over the world [43]. Worldwide this is a critical topic in line with the One Health issue [44]. Monitoring on the application and more important efficacy of antimicrobials regarding bacterial infections of farm animals is possible on principle in industrial countries. However, it is costly and difficult to standardize [36]. Published data from Canada in 2018 revealed a 51.6% susceptibility rate of 489 *E. coli* against trimethoprim-sulfamethoxazole, which is in consensus with our data (50%) (Figure 1 and Figure 2) [45]. Tetracycline was accounted to be effective in 36.8% and resembles our findings at 38.1% (Figure 1) [45]. Further, authors from the United States and Germany determined similar high resistance rates for tetracycline, with 71.1% and 70.9%. These data rather resemble the rate of 61.4% revealed in the present study (Figure 1) [46,47].

The antimicrobial class of fluoroquinolones includes enrofloxacin which is one of the substances of choice for the treatment of diarrhea in young cattle [29,48]. In Germany, the usage of fluoroquinolones has risen from 2011 to 2013 in human and veterinary medicine. This trend needs close monitoring to preserve the efficacy of the agent [27]. Fluoroquinolones are assessed as highest priority clinically important antimicrobials and as one of the few options for the treatment of serious *Salmonella* and *E. coli* infections in children recommended by the World Health Organization (WHO) [49]. The legislation reacted and passed a law in 2017 including obligatory antimicrobial susceptibility testing in case of the application of fluoroquinolones or third or fourth generation cephalosporines in Germany [50]. In the present study, the investigated *E. coli* isolated revealed a resistance rate of 34.1% regarding enrofloxacin (Figure 1). This finding correlates with published results from South America in 2017, with 36.4% [51].

Antimicrobial substances or closely related compounds may likewise be licensed for the use in man and animals. The application in an organism does trigger the development of antimicrobial resistance in present bacteria [49]. Legal restrictions regarding the use of cephalosporines, especially from the third and fourth generation, aim at a high prioritization of critically important antimicrobials in human medicine [49]. This is again in accordance with the terms of One Health [27,44]. The use of cephalosporines for the therapy of *E. coli* diarrhea in calves is a malpractice, as the effective therapeutic concentration is not reached within the gut [29]. Nonetheless, cephalosporin is the fifth-most commonly prescribed antimicrobial in the case of diarrhea with 8% according to a recent survey in Europe [30]. Regarding the third generation cephalosporine ceftiofur, a susceptibility rate of 86.4% could be determined in a study from Canada between 1994 and 2013 [45]. Significantly, our findings revealed 76.8% (Figure 1). Compared to data from the USA collected within the years 1960 until 2002 and in 2007, the resistance rate was at 7.4% and 11%, whereas in the present study the resistance rate of ceftiofur revealed 20.4% (Figure 1) [46,52]. This result is concerning, and the use of ceftiofur must be scrutinized critically, if not avoided completely. The resistance rates of the first generation cephalosporine, cephalothin, were lower in a comparable study regarding data within the period of 1960 to 2002, with 20.1%, in contrast to our results with an average rate at 46.1% from 2015 to 2019 (Figure 1) [46]. Currently, the standard antimicrobial therapy of mastitis in cows includes penicillins as well as first and second generation cephalosporines in the EU. Traces of antibiotics may reach the calves through the feeding of antibiotic contaminated waste milk [36]. To predict a reliable trend regarding the prevalence of ESBL-producing *E. coli*, PCR and sequencing methods should be applied to investigate the existence of ESBL- encoding genes as these are probably more accurate than the phenotypic characterization [53]. A study from 2013 revealed high rates (32.8%, 196 of 598 samples) of ESBL-encoding *E. coli* on dairy and beef cattle farms in Bavaria [54].

Completely inconsistent data are publicly available regarding the resistant rates for *E. coli* isolates and the substance florfenicol within the phenicol group. A 78% share of resistant isolates was determined in a study from the USA in 2006, only a 28% share from Canada in 2018, and a share of 35% from Bavaria, Germany, in 2002 [45,52,55]. In the present study, a rather higher resistance rate of 60.6% was determined for florfenicol (Figure 1). There was no information about ages of animals within the American and Canadian studies [45,52]. Since lower resistance rates were previously published in older animals for the substances ampicillin, tetracycline, streptomycin, sulfamethoxazole and chloramphenicol, this might accordingly apply for florfenicol [32]. This argument, however, still does not explain the diverse results of the Bavarian study from 2002 and the present study (Figure 1) [55].

With a 9% share of the most frequently listed antimicrobials, aminoglycosides remain at the fourth top position for the treatment of diarrhea in calves [30]. As these are almost solely used in the therapy of enterococcal endocarditis and multidrug-resistant tuberculosis in humans, they account to the high priority, clinically important antimicrobials in human medicine [49]. An application in veterinary medicine should therefore be prudent and well considered. Gentamicin belongs to the aminoglycoside antimicrobial class and has a withdrawal time for meat of more than 200 days in Germany for cattle and the indication of gastrointestinal disease. As this is economically hardly acceptable, the application of gentamicin is quite limited [48]. However, resistance to gentamicin among *E. coli* isolated from animals has been increasing from 0% to 40% between 1970 and 2002 within the United States [46]. Another long-term investigation from Germany revealed a further decrease of resistance rates including data from 2010 until 2013, and 2016 until 2017, respectively [47]. In the present study, the resistance rate of *E. coli* against Gentamicin was at 14.1% (Figure 1). Likewise, spectinomycin is an aminoglycoside antibiotic as well, and frequently used in combination with lincomycin for oral application in the treatment of simultaneous infection of the respiratory and the gastrointestinal tract in calves. The meat withdrawal time of 21 days is acceptable for farmers and practitioners and may be an explanation for the frequent prescription [48]. Within the present study and correspondingly a resistance rate of 48.9% was revealed in calves (Figure 1).

As stated by the WHO, the antimicrobial class of polymyxins accounts for the highest priority in critically important antimicrobials regarding the treatment of serious infections with Enterobacteriaceae and *Pseudomonas aeruginosa* in human medicine [49]. Despite rather frequent prescription of polymyxins in the treatment of diarrhea in animals, investigated *E. coli* isolates are still highly susceptible [30]. In the present study, the resistance rate against colistin revealed to be only 1.8% (Figure 1). Corresponding to this suggestion, another study revealed that only 3.8% of the isolates were resistant to colistin [47].

The aminopenicillin family, as well as the preparation amoxicillin-clavulanate, belong to the high priority critically important antimicrobials for the therapy of *Listeria* and *Enterococcus* spp. infections in humans according to the WHO [49]. For the aminopenicillin, ampicillin, an alarming resistance rate of 76.3% was determined in *E. coli* published in a most recent study from Germany [47]. Regrettably, a rate of 69.5% was determined in the present work as a similar result (Figure 1). Consequently, the recommendation on the usage of ampicillin for the treatment of calf diarrhea cannot further be continued. The amoxicillin-clavulanate susceptibility rate averaged at 57% in Germany in 2013 [27]. In the present study, the average susceptibility rate was 71.9%, and the resistance rate was 8.6% (Figure 1). Accordingly, a recently published study reported 7% of resistant *E. coli* isolates in Germany in 2018 [34]. Analogical to the report on the resistance monitoring study 2018 of the Federal Office of Consumer Protection and Food Safety, Germany, we determined decreasing non-susceptibility rates regarding the clinically important antimicrobial amoxicillin-clavulanate [34]. In conclusion, the resistance rates of *E. coli* against amoxicillin-clavulanate have decreased since 2013 and remained on a constant level within the years 2015 and 2019. This is a positive trend is beneficial for the One Health point of view [27].

Comparing data originating from other continents and collected over the last 60 years clearly reveals an increase of resistance regarding *E. coli* in nine out of the 12 tested drugs, namely gentamicin, cephalothin, ceftiofur, enrofloxacin, trimethoprim-sulfamethoxazole, ampicillin, amoxicillin-clavulanate, florfenicol and tetracycline [27,34,45,46,47,51,52,55]. Out of the 12 tested drugs in the present study, eight substances are similarly suitable for the treatment of human patients, namely gentamicin, spectinomycin, cephalothin, ampicillin, tetracycline, amoxicillin-clavulanate, colistin and trimethoprim-sulfamethoxazole (Figure 1) [49]. The application of these in veterinary medicine should be prudent due to the One Health aspect.

In a published study from Canada in 2018, 48.7% of multidrug-resistant *E. coli* were isolated from ruminants [45]. Within another study from the USA covering the years 1950 until 2002, a significantly increasing trend in resistance was observed for ampicillin, sulfonamide and tetracycline antibiotics regarding more than 1700 *E. coli* isolates. Two of these strains were identified as pandrug-resistant and originated from cattle in 2001 [46]. Further, multidrug resistance in *E. coli* increased from 7.2% to 63% between 1950 and 2002. Finally, 59.1% of the strains recovered form cattle were classified as multidrug resistant in the USA [46]. In the present study, we detected an even higher rate of 84.2% regarding multidrug resistance, 15.7% extensively drug-resistance and 0.1% pandrug-resistance (Figure 3). Furthermore, there were no exclusively susceptible isolates found amongst 108 isolates recovered in 2019 and 2020 from diarrheic calves in Bavaria (Appendix A). Comparably high levels of antimicrobial resistance were published regarding the countries Brazil and Uruguay. Calves aged up to 60 days revealed a multidrug-resistance rate in *E. coli* at 78.7%, and at 61.6%, respectively [51]. As published, these bacteria occurred frequently in herds with high levels of diarrhea symptoms and subsequent antimicrobial therapy, as equally described in the present study [31].

Besides antimicrobial resistance, the determination of virulence regarding infectious agents is crucial in diagnostics. The discrimination from commensal *E. coli* was determined investigating virulence factors and evaluating the pathogenicity of isolates. As published, the *E. coli* serotypes O139:K82, O8:K87 and O147:K89 are pathogenic in swine [6]. However, in the present study, a fair amount of such isolates, six out of 108, were isolated from cattle, respectively (Table 2, Appendix A). In laboratory diagnostics, implication of these serotypes should therefore be considered. Three isolates were identified as the serotype O78:K80, which frequently causes septicemia in calves (Table 2) [5,7,56]. However, more than one third, 38%, of the *E. coli* in this study revealed to be entirely seronegative (Table 3, Appendix A), as it was as well published previously [57]. Preferably and in accordance with the One Health approach, the screening of *E. coli* isolated from diseased animals should always be of interest to identify zoonotic and human pathogenic serotypes [25]. As a matter of fact, formula associated with severe human syndromes included the serotypes O26, O103, O111, O117, O128, O145 and O146 respectively [13,22,23,58].

In recent studies, the fimbrial adhesins F17, F41 and F5 were frequently and significantly correlated with diseased calves compared to healthy animals [4,9]. These findings clearly correspond to the results of the present study (Table 2, Appendix A). Other selective fimbrial antigens, F4, F6 and F18, occur frequently in isolates from diarrheic piglets [1,10,59]. As to be expected, we did not detect these amongst our strains isolated from calves (Appendix A). Even five serologically F4 positive isolates were not confirmed within our molecular investigation (Table 2, Appendix A). We assume that none of these isolates carry the specific primer sites, or agglutination was non-specific [9]. However, working at a federal state laboratory, we do research cross species infections especially among farm animals [60]. Furthermore, we consider the One Health approach, here especially the idea from farm to fork, and therefore continuously consider possible correlations between food-borne human pathogens and isolates from farm animals [27,44].

As published, hemolysis in *E. coli* isolates from piglets is a reliable diagnostic marker for virulence and pathogenicity [61,62,63]. Within the present study, only few (3/108) isolates revealed a hemolytic phenotype that was not even confirmed within the molecular analysis (Appendix A). We conclude that hemolysis is not a relevant marker for virulence of *E. coli* isolated from calves in the present study. This statement is in accordance with prior publications [64,65].

Regarding the present study, ST-I was found in similar prevalence at a rate of 6.5% (7/108) compared to published data (Table 2, Appendix A) [4,66]. The enterotoxins LT and ST-II were not detected in the present study (Appendix A) and this again resembles data of relevant previous studies [4,56]. Concluding published data, ETEC isolated from calves only produced ST-I, whereas ETEC isolated from pigs may encode varying combinations of the enterotoxins LT, ST-I and ST-II [11,67]. In the present study, the detection rate of stx1 was very low and stx2 as well as intimin were not detected at all among the diarrheic calves’ isolates (Table 2, Appendix A). This finding matches the results of previously published data to a high degree [9,51,68]. Obviously, the detection rate of Shiga toxins rose with the number of colonies isolated from each clinical sample, suggesting the selection of up to 35 colonies [69,70]. In the present investigation however, only up to three colonies were analyzed per clinical sample (Appendix A). Other published results suggested a positive correlation between animal age and the amount of Shiga toxin, supporting our findings including animals of young age [69,70,71]. Targeted infection studies with STEC led to severe disease and bloody diarrhea in neonatal calves, but more recent studies disproved this observation revealing a still controversial discussion [4,72,73,74].

### Limits of the Study

The antimicrobial susceptibility testing was carried out with a standard panel of antibiotics currently used in veterinary diagnostics in Germany. The results are therefore limited to substances only partially prescribed in human diagnostics and sometimes even in veterinary medicine regarding other countries of the world.

A thorough molecular investigation of single isolates is fairly time consuming and costly compared to the benefit that might be drawn from the results. In routine diagnostics, the molecular methods therefore can hardly be kept up.

## 4. Materials and Methods

### 4.1. Study Design and Bacterial Isolates

At the Bavarian Health and Food Safety Authority in Germany 7358 fecal samples of diseased or deceased calves with enteritis younger than six weeks of age were analyzed and included in the present study. Samples were collected between January 2015 and December 2019. Clinical symptoms ranged from low general condition, diarrhea, fever, sepsis and sudden death, respectively. A total of 8713 *E. coli* strains were isolated and confirmed through positive fluorescence on ECD agar (Merck Millipore, Burlington, MA, USA) and a positive Kovacs-Indole reaction (Merck Millipore, Burlington, MA, USA). All isolates were subject to antimicrobial resistance testing, further analysis and cryopreservation at the internal vaccine laboratory.

### 4.2. Antimicrobial Susceptibility Testing

Antimicrobial susceptibility testing was carried out according to the protocols published in CLSI VET01, 5th edition (Clinical and Laboratory Standards Institute, Wayne, PA, USA) [41]. Breakpoints were adopted from CLSI Vet01S, 5th edition, and national breakpoints for farm animals [41,42,75]. We used the microbroth dilution method on the following twelve different antimicrobial agents (antimicrobial class): Amoxicillin-clavulanic acid (betalactam combination agent), enrofloxacin (fluoroquinolone), Trimethoprim-sulfamethoxazole (folate pathway inhibitor), gentamicin and spectinomycin (aminoglycosides), cephalothin (cephalosporin I and II), ceftiofur (cephalosporin III and IV), ampicillin (penicillin), florfenicol (phenicol), colistin (polymyxin), tetracycline (tetracycline) and tulathromycin (macrolide). A commercially available set was used according to the manufacturer’s instructions (Micronaut-S, Grosstiere 4, Merlin, Bruker, Bornheim, Germany). The minimum inhibitory concentration (MIC) of each isolate and antibiotic substance was metered using a photometric plate reader system (Micronaut Scan and MCN6 software, Merlin/ Sifin, Bruker, Bornheim, Germany). Subsequently, the MIC value was reconciled with supplemented CLSI breakpoints, to categorize the respective *E. coli* isolate into “susceptible”, “intermediate” and “non-susceptible” for each antimicrobial substance tested [41,42,75,76]. *E. coli* ATCC 25922 was used as quality control strain [41].

### 4.3. Phenotypic Analysis and Serotyping

We deeper investigated a subset of 108 *E. coli* isolated in 2019 and 2020 originating from 66 diarrheic calves. The isolates were subcultured on Gassner agar (Oxoid Deutschland GmbH, Wesel, Germany) to differentiate specific colony morphology. The expression of potential virulent F5 fimbria was investigated by subculturing the isolates on pH 7.5 stabile, “minimum of casein” (Minca) agar (Sifin Diagnostics GmbH, Berlin, Germany) as previously published [76]. Finally, potential hemolytic properties of isolates were interpreted as described with subcultures on Columbia Sheep Blood Agar (Sifin Diagnostics GmbH, Berlin, Germany) [77]. Growth incubation was carried out for 18 to 24 h at 37 °C at all times. Serotyping for specific O-antigens was carried out using two polyvalent and 14 monovalent agglutination sera in a hierarchical approach according to the manufacturer’s instructions (Sifin Diagnostics GmbH Berlin, Germany) (Table 3). If an isolate showed a positive agglutination reaction with a polyvalent serum, but none with any correspondent monovalent or several reactions with various correspondent monovalent sera, it was categorized as untypeable. If an isolate showed no positive agglutination with any serum, it was categorized as seronegative.

### 4.4. Molecular Investigation

The molecular characterization of the *E. coli* isolates in the present study aimed at surface antigens, toxins and virulence factors. In all, 14 different target genes were of interest. Amongst were seven fimbrial genes F4, F5, F6, F17, F18, F41 and the outer membrane protein O157:H-. Further, two virulence genes were included, here adhesin intimin (eaeA), and enterohemolysin (ehxA). Finally, PCR targets coding for five toxins were screened, including heat-labile toxin (LT), heat-stabile toxin I (ST-I) and II (ST-II), Shiga toxin 1 (stx1) and stx2 (Table 4). Primer sequences were adopted from published protocols [9,39,40]. All 14 qPCR assays were performed applying a singleplex high resolution melting method, using AccuMelt HRM SuperMix (Quantabio, Beverly MA, USA) in 20 µL volumes according to the manufacturer’s instructions. DNA was extracted after thermolysis. The primers were added in a concentration of 0.2 µM each, and 3 µL of template DNA was used. Polymerase chain reaction assays were conducted on a Stratagene MX3000P device (Agilent Technologies, Waldbronn, Germany). The cycling protocol comprised an initial single denaturation step for 10 min at 95 °C, followed by 40 cycles of annealing and polymerization for 30 s at 60 °C and 10 s at 95 °C. After completing amplification, the melting curve analysis was performed. Specific melting temperatures were determined for each molecular target and all tested isolates. Reference strains were used as positive controls and kindly provided from Prof. R. Bauerfeind (Justus-Liebig-Universität, Gießen, Germany), and purchased from the German Collection of Microorganisms and Cell Cultures GmbH (DSMZ, Braunschweig, Germany) (Table 4).

### 4.5. Statistical Analysis

All statistical analyses were performed using the free software R Studio version 1.2.5033 (RStudio, Inc., Boston, MA, USA). Resistance trends of three clinically relevant antimicrobials amoxicillin-clavulanate, enrofloxacin and trimethoprim-sulfamethoxazole were evaluated by calculating a logistic regression model. The respective year was set as a continuous variable. The resulting odds ratio (OR) > 1 indicated an increased resistance trend, whereat an OR < 1 indicated a decreased antimicrobial resistance. The Wald test was used to determine the statistical significance of the year-antimicrobial trend. A value of *p* < 0.05 was considered significant (Table 1).

## 5. Conclusions

We conclude that an extensive monitoring, characterization and the analysis of antimicrobial resistance regarding enteritis causing *E. coli* is crucial to determine the currently raging serotypes, virulent genotypes and most important, the resistance situation. It is then possible to calculate reliable tendencies and prognoses from data collected over long terms in routine diagnostics. This is an important premise for objective and professional treatment recommendations regarding humans and animals within the scope of One Health. A further goal should be a slowdown of the increasing antimicrobial resistance situation that constitutes a global public health threat.

## Figures and Tables

**Figure 1 antibiotics-11-00023-f001:**
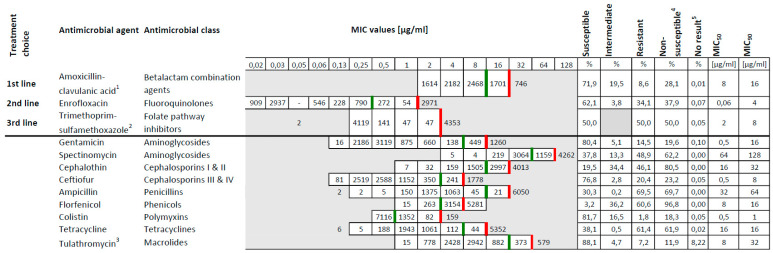
Minimum inhibitory concentration (MIC) distribution of 8713 *E. coli* isolates on 12 antimicrobial agents from 11 antimicrobial classes. The three first lines represent the clinically relevant substances, first to third treatment choices in buiatrics. The red line demarcates the breakpoint towards resistance, the green line a breakpoint towards intermediate. Regarding the two combination compounds, only the concentration of the former substance is presented; the ratio of amoxicillin:clavulanic acid is 2:1 (1), concentration ratio of trimethoprim:sulfamethoxazole is 1:19 (2). Tulathromycin has not been tested in the first quarter of 2015 (3). The summation of intermediate and resistant isolates was named non-susceptible (4). Some results were not evaluable (5).

**Figure 2 antibiotics-11-00023-f002:**
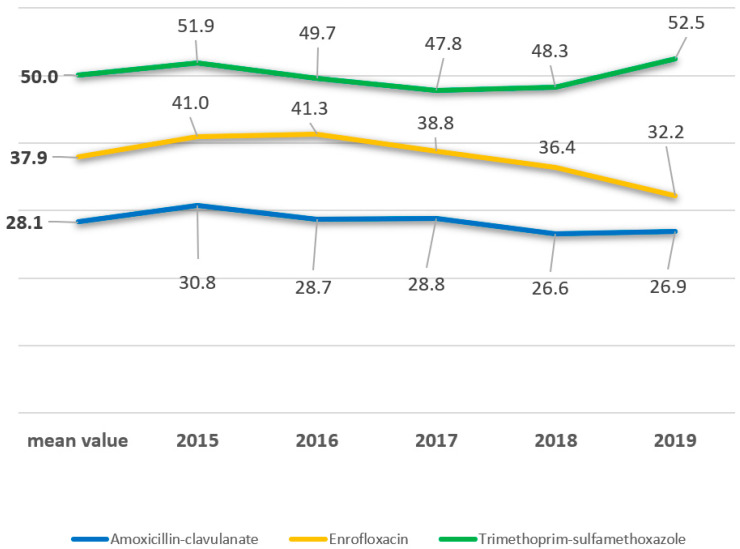
The mean value (bold) and the five-year trend on non-susceptible *E. coli* isolated from calves revealed the highest proportion of isolates against trimethoprim-sulfamethoxazole, followed by enrofloxacin and amoxicillin-clavulanate. The trends regarding enrofloxacin and amoxicillin-clavulanate remain at a stable level and rather tend towards a decrease regarding the number of non-susceptible isolates. The graph of non-susceptible isolates regarding trimethoprim-sulfamethoxazole reveals a decrease, 2016–2017, followed by a steep increase of non-susceptible isolates in 2019. The corresponding statistic parameters are presented in Table 1.

**Figure 3 antibiotics-11-00023-f003:**
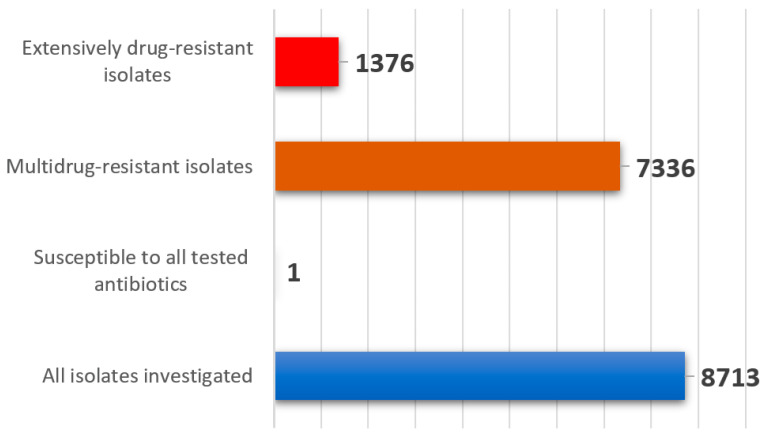
The classification of 8713 *E. coli* into extensively drug-resistant and multi drug-resistant isolates was carried out according to the expert proposal for standard definitions for acquired resistance. We categorized eight potential pandrug-resistant isolates in the category extensively drug resistant, as we only tested antimicrobials licensed for the veterinary use and did not include the latest antimicrobials available on the market.

**Table 1 antibiotics-11-00023-t001:** Statistic parameters regarding the increase or decrease of resistance values within the five-year period for the three clinically relevant antimicrobials (Figure 2).

Antimicrobial	Years	OR	CI (95%)
amoxicillin-clavulanate	2015–2019	0.95	0.92–0.98 ^1^
enrofloxacin	2015–2019	0.91	0.88–0.94 ^1^
	2015–2017	0.92	0.85–1.0 ^1^
trimethoprim-sulfamethoxazole	2015–2019	1.0	0.97–1.03
	2017–2019	1.11	1.03–1.19 ^1^

OR: odds ratio, CI: confidence interval, ^1^
*p*-value (Wald test) < 0.05.

**Table 2 antibiotics-11-00023-t002:** The serologic and molecular characterization revealed 13 different serotypes known to be pathogenic for cattle and other species. Furthermore, four different genotypes were detected with five different coding sequences for fimbria and/or toxins in one or more isolates. Some of the isolates were untypeable/ seronegative and did not reveal any of the investigated virulence factors (green box).

Serotype	Additionally Known for Pathogenicity in	Number of Isolates	Non-Virulent	Molecular Results			
				F17	F5ST-I	F5F41ST-I	stx1
O9:K35		6	5	1			
O9:K35/F5		1				1	
O101:K28		6	6				
O101:K28/F5		3			3		
O101:K30		1		1			
O101:K30/F5		3				3	
O101:K32		3	3				
O78:K80	Human/sheep	3	3				
O8:K87	Swine	3	3				
O139:K82	Swine	2	2				
O139:K82/F4	Swine	5	4	1			
O141:K85	Swine	1	1				
O147:K89	Swine	1		1			
untypeable		29	20	7			2
seronegative		41	37	4			
Total		108	84	15	3	4	2

**Table 3 antibiotics-11-00023-t003:** In all, 16 different polyvalent and monovalent (mono) antisera were used for the agglutination and the characterization of *E. coli*. The listed serotypes are known for their pathogenicity in humans and farm animals.

Antiserum for Initial Screening	Respective Follow Up Agglutination	Specific Serotypes Occur in Cattle, but Are Found as Well/Especially in
Polyvalent anti-*E. coli* C		
	O9:K35, mono	
	O101:K28, mono	
	O101:K30, mono	
	O101:K32, mono	
	F5, mono	
O78:K80, mono		Human, sheep
Polyvalent anti-*E. coli* P		Swine
	O8:K87, mono	
	O138:K81, mono	
	O139:K82, mono	
	O141:K85, mono	
	O147:K89, mono	
	O149:K91, mono	
	F4, mono	
O157:H7, mono		Association with food-poisoning

**Table 4 antibiotics-11-00023-t004:** Targets and primers for the molecular characterization of *E. coli* isolated from calves.

Target Protein	Gene(s)	Primer	Oligo Sequence (5’ -> 3’)	Size (bp)	Melting Temperature (°C) ± 0.2 °C	Reference	Reference Isolate
	F4	F4_F	GGTGGAACCAAACTGACCATTAC	102	81.0	[9]	7156
Fimbria/outer membrane protein		F4_R	TCCATCTACACCACCAGTTACTGG				
F5	F5_F	TTGGAAGCACCTTGCTTTAACC	101	77.4	[9]	7159
	F5_R	TCACTTGAGGGTATATGCGATCTTT				
F6	F6_F	GCGGATTAGCTCTTTCAGACCA	102	83.2	[9]	7155
	F6_R	TGACAGTACCGGCCGTAACTC				
F17	F17_F	ACTGAGGATTCTATGCRGAAAATTCAA	83	79.7	[9]	5397
	F17_R	CCGTCATAAGCAAGCGTAGCAG				
F18	F18_F	CCTGCTAAGCAAGAGAATATATCCAGA	82	73.3	[9]	7160
	F18_R	AGAACATATACTCAGTGCCAACAGAGAT				
F41	F41_F	CCTTTGTCATTTGGTGCGG	101	81.5	[9]	7159
	F41_R	TCAAATACTGTACCAGCAGAACCAC				
O157 (rfbE)	O157_F	CGATGAGTTTATCTGCAAGGTGAT	88	78.3	[39]	DSMZ 19206
	O157_R	TTTCACACTTATTGGATGGTCTCAA				
Adhesin	intimin (eaeA)	Intimin_F	CCAGCTTCAGTCGCGATCTC	91	86.1	[9]	7158
Intimin_R	GGCCTGCAACTGTGACGAA				
Hemolysin	enterohemolysin (ehxA)	ehec-F2	CGTTAAGGAACAGGAGGTGTCAGTA	142	79.5	[40]	DSMZ 19206
ehec-R	ATCATGTTTTCCGCCAATGAG				
Toxin	heat-labile toxin (LT)	LT_F	CTGCCATCGATTCCGTATATGAT	81	75.3	[9]	7157
LT_R	CAGAACTATGTTCGGAATATCGCA				
heat-stabile toxin (ST-I)	ST-I_F	TACCTCCCGTCATGTTGTTTCAC	101	76.1	[9]	7155
ST-I_R	CCTCGACATATAACATGATGCAACTC				
heat-stabile toxin (ST-II)	St-II_F	TTTTTCTATTGCTACAAATGCCTATGC	101	75.9	[9]	7156
St-II_R	AACCTTTTTTACAACTTTCCTTGGC				
Shiga toxin 1 (stx1)	Stx1_F	TCCCCAGTTCAATGTAAGATCAAC	81	79.0	[9]	7158
Stx1_R	TTTCGTACAACACTGGATGATCTCA				
Shiga toxin 2 (stx2)	Stx2_F	GAGTGACGACTGATTTGCATTCC	82	84.6	[9]	7158
Stx2_R	CCATGACAACGGACAGCAGTT

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
