# Peer review of "Antimicrobial Resistance, Serologic and Molecular Characterization of *E. coli* Isolated from Calves with Severe or Fatal Enteritis in Bavaria, Germany"

_antibiotics, 2021, doi:10.3390/antibiotics11010023_

Round 1
Reviewer 1 Report
The manuscript entitled “Antimicrobial resistance, serologic and molecular characterization of E. coli isolated from calves with severe or fatal enteritis” by Feuerstein et al. performed antimicrobial susceptibility testing, serologic and molecular characterization of E. coli isolates from diseased or deceased diarrheic calves to provide an updated information on the prevalence of virulence factors and antimicrobial resistance in animals in Bavaria, Germany. Interestingly they found substantially high rates of multi-drug resistance including pandrug-resistant isolates. This looks an interesting study that can be published. My specific comments are as following.
- The author mentioned that they used R studio for statistical analysis, it is not much clear what specific statistical test and R packages they used for significance analysis. Therefore, authors are suggested to mention the detail of statistical test performed in the manuscript.
- In general, the quality and visualization of presented figures can be improved.
- There seems some typing mistakes in the manuscript that needs to be corrected before publication.
Reviewer 2 Report
Dear Authot,
My sugestions are short.
- Please improve the tables and figures, specially the first table and the figures. ALso, some data should be better to show as a figure instead table, for example table 2.
- Review the manuscript, some litle details
- Discussion and conclusio. The author is talking under the ONE HEALTH concept, so then, The author should comment or associate the resistances with the "human" resistances in the área, and to discuss aot the possibel transmission or relation.
Thanks
Reviewer 3 Report
In the manuscript authors described the results of a survey on E coli recovered from calves reared in Bavaria, focusing the attention on antibiotic resistance and the assessment of some virulence factors.
The manuscript is well written but, for publication, some aspects should be clarified and some changes should be made.
The study was carried out in five years period and exclusivelly in a well-defined Land (the Bavaria), this should be reported in the title.
In the chapters "introduction" and "discussion" there are some references to pigs, piglets and humans; are they really useful for research aims? Given the research aims, the importance of the correlation of the data from calves, human and piglets (Table 3) should be better clarified. This relevance is absolutely not highlighted either in the title or in the Keywords
In the "results" the antibiotics in some cases are written with the full name (lines 102-104) in other with the abbreviation (lines 108-110); furthermore the manuscript lacks in the list of the 12 molecules considered, as well the relative abbreviations. Figure 1 lists antibiotics, but not the abbreviations.
In the chapter 2.2 (serological characterization) the data reported seems not to be the same reported in table 2. Please revise the text or the table.
No details on feature of cattle breeding in Bavaria is reported. Furthermore, it is not possible to understand if the sample taken into consideration (7,358 fecal samples; 8,713 E coli strains etc) has a statistical significance.
In chapter 4 (molecular investigation) and in table 4, more target genes searched than those considered in the results chapter are stated. Even if some targets have never been detected this should be reported, evaluated and discussed.
The conclusions chapter is too generic; it does not refer to the study results. In this chapter the importance of the data highlighted in this specific research must be emphasized.
Figure 1: it display a lot of data, probably too many. I suggest reporting the MIC results separately from those on % susceptibility (etc) to the 12 antibiotics.
Figure 3: I suggest to replace "none of these criteria" with "susceptible to all antibiotics tested", as reported in the text.
Table 2 : the data reported seems not to be the same reported in the text, please verify.
Reviewer 4 Report
The manuscript is well written and suitable for English with a clear structure. The authors worked on the prevalence of E. coli, virulence factors, serotypes, and trends in antimicrobial resistance. The study presents a fascinating piece of work. The study reported a magnitude of 8,713 E. coli isolates obtained from 7,358 samples over five years. The study requires only a few minor corrections.
The study reported 0.1% of pandrug-resistant strains using 12 antibiotics from the 11 antimicrobial categories. To report pandrug-resistant strains, the author should have included several other antibiotics from the same categories. Moreover, there are a number of new antimicrobial agents which were not used in the study. I recommend merging 0.1% of pandrug-resistant strains with the extensively drug-resistant strains.
The study design should be mentioned under heading 4.1.
The CLSI references in section 1.2 are from 2018, while the study was conducted from 2015 to 2019. If the authors performed AST in 2019, they can update this with CLSI 2019 reference or mention multiple CLSI references according to the year of use.
The study limitations should be included at the end of the discussion, such as some new antimicrobial agents were not used.
Conclusion: I have main concern on the conclusion, which is focused on future monitoring and directions for the surveillance. The authors should describe a conclusion based on their work.
Round 2
Reviewer 2 Report
Dear authors,
My suggestions and task have been not reached, yet.
I feel sorry to reject the manuscript in the present from.
Reviewer 3 Report
The manuscript has been revised as requested, I now consider it suitable for publication
Overall, two reviewers and two independent editors recommended acceptance of this manuscript.